# Simplified antibiotic regimens for young infants with possible serious bacterial infection when the referral is not feasible in the Democratic Republic of the Congo

**Adrien Lokangaka**[1]*, **Daniel Ishoso**[1], **Antoinette Tshefu**[1], **Michel Kalonji**[1], **Paulin Takoy**[1], **Jack Kokolomami**[1], **John Otomba**[1,2], **Samira Aboubaker**[3], **Shamim Ahmad Qazi**[3], **Yasir Bin Nisar**[4], **Rajiv Bahl**[4], **Carl Bose**[5], **Yves Coppieters**[6]

1 Faculté de Médecine, Kinshasa School of Public Health, Université de Kinshasa, Kinshasa, Democratic Republic of the Congo, 2 World Health Organization, Kinshasa, Democratic Republic of the Congo, 3 World Health Organization (Consultants), Geneva, Switzerland, 4 Department of Maternal, Newborn, Child and Adolescent Health and Ageing, World Health Organization, Geneva, Switzerland, 5 University of North Carolina at Chapel Hill, Chapel Hill, North Carolina, United States of America, 6 School of Public Health, Université Libre de Bruxelles, Brussels, Belgium

* adrinloks@gmail.com

## Abstract

### Introduction

Neonates with serious bacterial infections should be treated with injectable antibiotics after hospitalization, which may not be feasible in many low resource settings. In 2015, the World Health Organization (WHO) launched a guideline for the management of young infants (0–59 days old) with possible serious bacterial infection (PSBI) when referral for hospital treatment is not feasible. We evaluated the feasibility of the WHO guideline implementation in the Democratic Republic of the Congo (DRC) to achieve high coverage of PSBI treatment.

### Methods

From April 2016 to March 2017, in a longitudinal, descriptive, mixed methods implementation research study, we implemented WHO PSBI guideline for sick young infants (0–59 dyas of age) in the public health programme setting in five health areas of North and South Ubangi Provinces with an overall population of about 60,000. We conducted policy dialogue with national and sub-national level government planners, decision-makers, academics and other stakeholders. We established a Technical Support Unit to provide implementation support. We built the capacity of health workers and managers and ensured the availability of necessary medicines and commodities. We followed infants with PSBI signs up to 14 days. The research team systematically collected data on adherence to treatment and outcomes.

### Results

We identified 3050 live births and 285 (9.3%) young infants with signs of PSBI in the study area, of whom 256 were treated. Published data have reported 10% PSBI incidence rate in

**Data Availability Statement:** All relevant data are within the paper and its Supporting information files.

**Funding:** The study was funded by the Bill and Melinda Gates Foundation through a grant to the World Health Organization. The funder had no role in study design, data collection and analysis, decision to publish, or preparation of the manuscript.

**Competing interests:** Rajiv Bahl and Yasir Bin Nisar are staff members of the World Health Organization. The expressed views and opinions do not necessarily express the policies of the World Health Organization. This does not alter our adherence to PLOS ONE policies on sharing data and materials.

young infants. Therefore, the estimated coverage of treatment was 83.9% (256/305). Another 426 from outside the study catchment area were also identified with PSBI signs by the nurses of a health centre within the study area. Thus, a total of 711 young infants with PSBI were identified, 285 (40%) 7–59 days old infants had fast breathing (pneumonia), 141 (20%) 0–6 days old had fast breathing (severe pneumonia), 233 (33%) had signs of clinical severe infection (CSI), and 52 (7%) had signs of critical illness. Referral to a hospital was advised to 426 (60%) infants with CSI, critical illness or severe pneumonia. The referral was refused by 282 families who accepted simplified antibiotic treatment on an outpatient basis at the health centres. Treatment failure among those who received outpatient treatment occurred in 10/128 (8%) with severe pneumonia, 25/147 (17%) with CSI, including one death, and 2/7 (29%) young infants with a critical illness. Among 285 infants with pneumonia, 257 (90%) received oral amoxicillin treatment, and 8 (3%) failed treatment. Adherence to outpatient treatment was 98% to 100% for various PSBI sub-categories. Among 144 infants treated in a hospital, 8% (1/13) with severe pneumonia, 23% (20/86) with CSI and 40% (18/45) with critical illness died.

## Conclusion

Implementation of the WHO PSBI guideline when a referral was not possible was feasible in our context with high coverage. Without financial and technical input to strengthen the health system at all levels, including the community and the referral level, it may not be possible to achieve and sustain the same high treatment coverage.

## Introduction

An estimated 2.5 million neonates died in 2018, accounting for about 47% of all deaths among children under five years of age [1,2]. Though global neonatal mortality decreased from 31 deaths per 1000 live births in 2000 to 19 per 1000 live births in 2015, the higher rate of 29 per 1000 observed in Central and Southern Asia and sub-Saharan Africa remains a concern [3,4]. While a large number of neonatal deaths were related to prematurity and birth asphyxia, a substantial number were due to serious bacterial infections such as neonatal sepsis and pneumonia [2,5]. The ANISA study from South-East Asia reported that among babies with possible serious bacterial infection (PSBI) who died with an attributable cause, 92% were bacterial [6]. In the Democratic Republic of the Congo (DRC), the neonatal mortality rate was estimated to be 29 per 1000 live births in 2018 [7].

The World Health Organization (WHO) recommends that infants with signs of a serious bacterial infection or neonatal sepsis be referred to a hospital and treated with injectable antibiotics, as it is estimated that most deaths are bacterial in origin [8]. Referral in low-resource settings may not be feasible or may not be accepted by families because of the cost of care in a hospital, lack of available or suitable transportation to a hospital, lack of hospital services, perceived low quality of care, or cultural beliefs about the inevitable mortality among hospitalized infants [9–13]. Studies from Southeast Asia have shown that when referral is not possible, sick young infants can be treated in the community or on an outpatient basis [14–16]. These studies were followed by randomized controlled trials comparing various simplified antibiotic regimens conducted in Bangladesh, the DRC, Kenya, Nigeria and Pakistan, which confirmed the safety and efficacy of outpatient treatment for these infants [17–20]. Using this evidence,

WHO developed a guideline for managing PSBI in sick young infants when referral was not feasible [21]. The implementation of this guideline should result in increasing access to treatment for more sick young infants, which could result in neonatal mortality reduction.

Before the implementation and scale-up of this guideline in the DRC, public health officials required a test of the feasibility and practicality of implementing the WHO guideline within the local programme context. This paper describes the feasibility and implementation of the WHO PSBI management guideline by the health authorities in one region of the DRC. Our objectives were: i) all health facilities provide management of PSBI; ii) at least 80% of all the expected number of infants with PSBI to be identified and treated; and iii) among treated infants, at least 80% should receive adequate treatment.

## Methods

### Study site

The study was conducted in rural areas of North and South Ubangi provinces in the northern region of the DRC, which included five health areas serving a population of about 60,000. These areas were selected because they are typical of many rural regions of the DRC, and we have an established health research infrastructure in this region. Health system details are given in Panel 1.

### Panel 1—Health system information

Operationally, primary health care is provided in catchment areas or health zones. Between 100,000–250,000 inhabitants reside in each health zone; each zone has one general referral hospital. Health zones are typically divided into 8–15 health areas, each with one health centre covering a population of 5000–10 000. All primary health care in the study area is provided through health centres, the points where skilled health providers and the community intersect, and treatment and health promotion activities are conducted. Each health centre is staffed by at least one trained nurse, who provides services and oversees the activities of community health workers (CHWs). CHWs are literate village volunteers who engage primarily in health promotion activities and work within the government system. They are unpaid; the only incentive they receive is free health care for themselves and their families.

Patients typically self-refer to health centres, which are around 1 to 8 kilometres away from communities. Those who require inpatient or speciality care are referred to nearby hospitals which are usually managed by general physicians. Distances from health centres to hospitals vary from one to over 50 kilometres. Most people travel by foot; few individuals own bicycles. Privately-owned motorbikes and cars are virtually non-existent, although occasionally, commercial trucks will transport passengers to larger towns. Patients who need access to hospital care and are unable to walk are usually placed on a stretcher and carried. It can take up to 10 hours for a patient to travel to a hospital from a rural health centre.

### Study design

This was a longitudinal, descriptive, mixed methods implementation research study, which evaluated the feasibility of WHO PSBI management guideline implementation when the

referral was not feasible at first-level health facilities, i.e. health centres in a programme setting to increase coverage of treatment. We report the details of the implementation strategy, the coverage of PSBI treatment, the rates of referral and refusal to accept a referral for inpatient care, the rates of implementation of the simplified antibiotic regimens on an outpatient basis when the referral was not accepted and the outcomes of care.

## Policy dialogue phase

In February 2015, a national orientation and policy dialogue meeting was organized by the DRC Ministry of Health (MOH) in collaboration with WHO. Participants included pro-gramme managers from newborn- and child health-related programmes, MOH Technical Advisory Group members, international partners (UNICEF, Save the Children, United States Agency for International Development), and academic institutions. At that meeting, the lack of a standard protocol for treating young infants with PSBI when a referral is not feasible, high newborn mortality, challenges of referral in rural areas and the need for action were discussed along with the results of the African Neonatal Sepsis Trial (AFRINEST) [19,20]. Table 1 sum-marizes the recommendations of the participants and agreements reached during the meeting. Recommendations regarding treatment were consistent with the WHO guidelines.

A second policy dialogue meeting was held at the provincial level to provide additional ori-entation and ensure buy-in from local authorities. The Provincial Minister of Health and the Directors of the Division of Health of North and South Ubangi Provinces and members of a

**Table 1. Decisions made during the policy dialogue meeting for the management of sick young infants with possible serious bacterial infection (PSBI)\* when a referral is not feasible.**

| Policy question | Decision |
|---|---|
| Who will identify sick young infants in the community? | Community Health Workers (CHWs), families |
| Where will sick young infants be assessed? | Health centres |
| Who will assess and confirm classification? | Health centre nurses |
| Who will provide treatment if a referral to the hospital is not accepted by the family? | Health centre nurses |
| Where will this treatment be provided? | At the health centre and, if needed, assess/treat at home |
| What treatment regimen will be given to 7–59 days old young infants with fast breathing† only (pneumonia)? | Oral amoxicillin twice daily for 7 days on an outpatient basis without referral to hospital. Mandatory visit on day 4 |
| Young infants 0–6 days old with fast breathing† only (severe pneumonia) will be referred. If a referral is refused, what treatment regimen will be given? | Oral amoxicillin twice daily for 7 days on an outpatient basis. Mandatory visit on day 4 |
| Young infants with signs of clinical severe infection (CSI) ‡ will be urgently referred to a hospital. If a referral to hospital is refused, what treatment regimen will be given to them? | Injection of gentamicin once daily for 2 days plus oral amoxicillin twice daily for 7 days on an outpatient basis. Mandatory follow-up visit on day 4 |
| Critically ill⫫ young infants will be urgently referred. If a referral is refused, what treatment regimen will be given to them? | Gentamicin injection once daily plus ampicillin injection twice daily for 7 days on an outpatient basis |
| Where should the early implementation sites be? | Karawa and Bominenge health zones |

\* PSBI—defined as the presence of any one of the following signs: i) not able to feed since birth, stopped feeding well or not feeding at all; ii) convulsions; iii) severe chest indrawing; iv) high body temperature ($\geq$ 38˚C); v) low body temperature ($<$ 35.5˚C); vi) movement only when stimulated or no movement at all; and vii) fast breathing (60 breaths per minute or more).

† Fast breathing was defined as a respiratory rate of 60 or more breaths per minute.

‡ Clinical severe infection (CSI)–defined as the presence of any one of the following signs: i) severe chest indrawing; ii) high body temperature ($\geq$ 38˚C); iii) low body temperature ($<$ 35.5˚C); iv) movement only when stimulated; v) stopped feeding well.

⫫ Critically ill—defined as any one of the following signs: i) not able to feed at all; ii) convulsions or fits; iii) no movement at all; iv) unable to cry; v) cyanosis; vi) bulging fontanelle.

Technical Support Unit (TSU) attended this meeting. The group decided to implement the PSBI guideline in five health areas selected as a convenience sample for early learning.

### Preparatory phase

A TSU was set up shortly after the initial policy dialogue meeting. It consisted of experts from the Kinshasa School of Public Health, a paediatrician from the Department of Paediatrics of the University of Kinshasa teaching hospital and the MOH responsible officer for neonatal health. The role of the TSU was to build the capacity of health care providers in PSBI management, to provide necessary technical assistance for the implementation of the intervention and to assist in the implementation research. The responsibilities of the TSU and the Health Zone Directors are described in Panel 2.

---

#### Panel 2. Responsibilities of the Technical Support Unit (TSU) and Health Zone Directors

##### Responsibilities of the Technical Support Unit (TSU)

- Adapt the generic WHO protocol, WHO/UNICEF Integrated Management of Childhood Illness (IMCI) chart booklet [22], developing the manual of procedures, study tools and data collection instruments and training materials

- Training community health workers, health centres nurses, and public health programme supervisors in case management, supervision, monitoring and documentation

- Training study staff in all necessary materials related to the implementation research

- Facilitating individual referral from health centres to the hospitals

- Ensuring the availability of commodities and supplies

- Refining the implementation guide in the light of field experience

- Provide technical support and quality control for implementation

##### Responsibilities of the Health Zone Directors

- Ensure that all health centres were adequately staffed according to the Ministry of Health (MOH) guideline

- Ensure that all health centre nurses were properly trained in the WHO possible serious bacterial infection (PSBI) guideline, IMCI and related training materials [21,22] and were competent to provide outpatient treatment to sick young infants

- Regularly monitor implementation of the WHO PSBI guideline in all health centres

- Identify areas where improvement is needed and together with the TSU find solutions-Serve as the link between the MOH, TSU, health centres and the study communities

---

## Situation analysis of the study area

A situation analysis of the study area was conducted to determine the readiness of the health facilities for PSBI management according to the WHO guideline. The key findings included: 1) stock-outs of medicines for the treatment of sick young infants with PSBI (especially oral amoxicillin) was common; 2) cost of health care services was beyond the means of most of the population; and 3) health centres lacked essential supplies/equipment (thermometers, weighing scales, etc.). The condition of the two referral district hospitals in the study area reflected the continuing economic crisis of the country. Each has four wards (internal medicine, obstetrics and gynaecology, pediatrics and surgery). Many of the buildings were in poor condition, with inadequate water and sanitation and electric supply. Usually, inpatient treatments were provided only when families could afford to pay for them. Hospitals were severely under-staffed. Blood bank services, essential equipment like incubators, infant warmers and medical supplies were inadequate. Laboratory capacity was limited with no microbiology facilities. Even oxygen was unavailable most of the time.

Private health providers are extremely rare or non-existent in this area. Traditional health providers exist, but their influence is steadily being decreased due to restrictions by government authorities.

CHWs conducted a census in all villages of the study areas to identify pregnant women and recent births.

## Barriers to health care seeking

Prior to implementation, we used qualitative methods to identify barriers to health care seeking. We held 20 focus groups with mothers, fathers, traditional birth attendants, and CHWs and 15 in-depth interviews with opinion leaders (physicians, political and administrative authorities, nurses and religious leaders). In each FGD 8 to 10 people participated. The discussion took place at the health center where people were convened. The selection of participants for the FGDs was guided by the health center nurses and the information provided by the CHWs. The criteria of homogeneity of residence, age, marital status, and level of education were also taken into account. These interviews were conducted under the guidance of two study physicians and two anthropologists from the University of Kinshasa. A total of 220 individuals, including 110 women and 95 men, participated in the FGDs, and 15 men in the in-depth interviews. The interviews recorded in Lingala and Ngbaka were translated into French and transcribed before being grouped and coded by theme and analyzed using ATLAS.ti version 7 software.

Our implementation strategy was informed by the major themes that emerged from the focus group discussions (Table 2).

## Building health system capacity

**Training and refresher training.** Two master trainers from DRC attended a master trainers' workshop organized by the WHO in collaboration with the University College Hospital of Ibadan, Nigeria. The three-day clinical training took place in Ibadan, which was followed by a three-day workshop to review and discuss the study-specific procedures. The master trainers received training on the integrated management of childhood illness (IMCI) [22], which included the WHO PSBI management guideline component [21] and the manual of operations for the implementation strategy. They were also trained on data collection tools, the consent process and documentation. Using a similar curriculum, the DRC master trainers subsequently trained nurses at the health centres in the study area at a six-day workshop. All trainees had to pass the clinical standardization assessment exercises at the end of the initial

**Table 2. Major themes of the barriers to healthcare seeking for young infants with PSBI identified during focus group discussions and interviews.**

| Levels | Barriers |
|---|---|
| System level | • Poor accessibility due to long distance and non-practical roads<br>• Monetization of health services<br>• Frequent stock-outs of medicine and supplies |
| Health providers' level | • Monetization of health care providers (no money, no care)<br>• Poor reception<br>• Long waiting time before consultation<br>• Refusal to change treatment even if the health condition does not improve |
| Family level | • Financial constraints (poverty)<br>• Cultural beliefs (diseases are caused by witchcraft, demons, etc.)<br>• Lack of transportation means (bikes, motorbikes, etc.) |

and refresher training courses to be certified for the project. Pre- and post-tests were carried out, which were organized by the MOH in collaboration with the TSU.

CHWs in the study area received six-day training on 'Managing the Newborn at Home' using the WHO/UNICEF training modules on home care of newborns [23]. The modules highlighted community interventions that were crucial to reduce newborn mortality. This training course aimed to equip the CHWs with skills needed to conduct effective home visits (i.e. communication, building good relationships with the family, stressing the importance of antenatal care for pregnant women, etc.). CHWs also developed competencies in assessing breastfeeding, danger signs, and measuring the weight of a newborn. They were taught to refer or provide home care depending on the results of the assessment.

**Job aids.** Training materials and job aids were developed for the health centre staff and CHWs and made available to support implementation. They included the young infant IMCI chart booklet, young infant IMNCI facilitator guide, young infant IMNCI participant manual, counselling cards for mother and baby cards in the local language.

**Commodities and financial subsidies for care.** Funding was provided to the selected health centres to ensure the availability of medicine and other supplies, including oral amoxicillin dispersible tablets, injectable gentamicin and injectable ampicillin, syringes, alcohol, swabs and registers. Management Sciences for Health, an implementation partner, supplied respiratory rate counting timers, infant weighing scales and digital thermometers. The requirement for various commodities was estimated based on the expected number of PSBI cases to be seen at the health centres. The quantity of medicines purchased was double the estimated amount to avoid stock-outs as it was expected that when communities came to know of free treatment for sick young infants, families from surrounding areas (outside of the study catchment area) would start bringing them to seek care at the study health centres. We provided bicycles to the Health Centre nurses and CHWs to facilitate home visits and transport mothers and their sick babies to the nearest health centre or the hospital if needed. The bicycles had long seats to comfortably transport patients and their mothers. We provided free health services for newborns and young infants, and CHWs and health centre nurses received a stipend for the additional tasks. Nurses received USD 50, and CHWs received USD 20 per month from the study.

## Implementation phase

**Supervision.** Supervision was ensured at different levels. At the central level, the government programme manager supervised the provincial division health officers, the health zone

directors and the health centre nurses every quarter to ensure smooth implementation. The programme manager verified the number of supervision activities carried out by the health zone directors, identified implementation challenges, provided feedback to the health centre and study staff and contributed to solving challenges met. The provincial division health officers visited the health zone once a month to review implementation progress. The health zone directors and health zone supervisors supervised health centre nurses and CHWs on a biweekly basis and validated their assessments. They performed joint home visits and assessed infants for danger signs to make sure study procedures were followed correctly. TSU members also joined some of the supervisory visits and provided on-the-job skills reinforcement.

**Community surveillance and identification of sick young infants.** After the preparatory phase, when CHWs had conducted a census, other methods were also used to identify pregnancies and births, including self-reporting to a CHW, identification of pregnant women at antenatal clinics in community health facilities, and referrals from traditional birth attendants or other key informants.

As part of the implementation strategy, pregnant women were visited twice during pregnancy. The first visit occurred as soon as possible after the pregnancy was confirmed. During that visit, the CHW encouraged the woman to seek formal antenatal care at health centres as early as possible and promoted the benefits of giving birth in a health facility. The second visit occurred about two months before the estimated date of delivery. The objective of the visit was to review birth plans and encourage the family to follow optimal newborn care practices after birth. CHWs conducted postnatal visits on days 1, 3 and 7. Additional home visits on days 2 and 14 were conducted for infants weighing less than 2.5 kg at birth.

During the postnatal home visits, CHWs provided advice regarding newborn care; nurses provided this advice during visits to the health centres. This advice adhered to recommendations outlined in the WHO/UNICEF Guideline on home care of newborns [23], including exclusive breastfeeding promotion, keeping the baby warm, hygienic practices and vaccination. Mothers were also counselled on recognition of danger signs. Young infants who exhibited any danger sign suggestive of PSBI identified by a CHW or mother were referred to a health centre for evaluation. At the health centre, the nurse assessed the sick young infants to determine if a PSBI existed. In a few cases, the health centre nurses visited the villages to confirm signs of PSBI when the CHW identified sick young infants.

**Referral of infants with signs of PSBI.** Young infants 7–59 days old with only fast breathing were treated with an oral antibiotic (amoxicillin) without a referral, whereas those infants with signs of severe pneumonia, CSI, or critical illness were referred to a hospital for appropriate treatment (Table 1). The health centre nurse encouraged and attempted to work with the families to overcome barriers to referral identified by them, e.g., fear of hospitalization, distance to the facility and lack of transportation. For those who accepted the referral, the health centre nurse provided pre-referral treatment (first injection of gentamicin and dose of oral amoxicillin). Referral to the hospital was facilitated and CHWs would take the sick infant and mother to the hospital on a bicycle if other transport was unavailable. If families declined referral, infants with CSI were offered a simplified antibiotic regimen of injectable gentamicin and oral amoxicillin or only oral amoxicillin for infants with severe pneumonia (Table 1).

**Administration of treatment.** Injections and the first daily dose of oral amoxicillin were administered by the health centre nurse, while the second oral dose for the day was administered by a family member. Dose details of antibiotics are given in S1 and S2 Tables. Those initiated on injectable therapy were seen again on subsequent days at the health centres for delivering injections. The family was asked to bring the infant to the health facility to complete the subsequent injection course. The health centre nurses live near health centres. On the

weekend or official holiday, an arrangement was made to provide the injectable therapy for sick young infants who needed it.

*Adequate treatment* was defined as two injections of gentamicin and at least 10 of the 14 oral amoxicillin doses.

**Monitoring and follow-up of patients on treatment.** All young infants receiving antibiotic treatment were evaluated periodically by health centre nurses and CHWs. During the second day of injection, the young infant was assessed for danger signs. At home, CHWs visited the family to promote home care practices, review progress and remind the family of the scheduled follow-up visits to the health centre. Families were counselled on signs of deterioration and were asked to bring the young infant back to the CHW or health centre if these signs appeared. If the child did not improve or deteriorated, the health centre nurse would insist on a referral. Young infants classified as having fast breathing pneumonia and others treated on an outpatient basis were asked to come to the health centre on day 4 for mandatory follow-up and then again on day 8 of treatment.

**Quality assurance.** The health zone director, study coordinator and nursing supervisors assessed the quality of implementation of surveillance and treatment during periodic visits to health centres. They observed specific activities performed by health centre nurses and CHWs, including counselling of pregnant women; screening for danger signs; provision of antibiotic treatment; and follow-up. Case report forms were reviewed for accuracy and completeness. Feedback was provided to the nurses and CHWs immediately and/or during monthly meetings. Each quarter, the TSU conducted standardization exercises with nurses and CHWs focusing on counting the breath, assessing infants for severe chest indrawing and other signs of PSBI, and measuring body temperatures.

**Data collection.** Quantitative data collected by the CHWs included pregnancy outcomes, other pregnancy-related information (place of birth, birth attendant, etc.), and danger signs observed during postnatal visits. The information was recorded on registers and study-specific data forms. One study nurse for each health centre was hired and paid through study funds specifically for this implementation research. The study nurses collected data describing signs and classification among young infants presenting at the health centre with possible infection. They were present when health centre nurses assessed young infants for PSBI signs, assisted health centre nurses with the consent process and recorded data on study forms, including treatment and follow-up information.

Qualitative data were also collected to assess the challenges and solutions during implementation. We conducted key informant interviews with village leaders. We also carried out focus group discussions with health workers and caregivers/families of sick young infants.

## Data processing and analysis

A study clerk entered data using CS-Pro software. After cleaning the data, checking for missing, outlier and invalid values, the analysis was conducted using Stata version 12.0 (Stata Corp, College Station, TX). Descriptive analysis was conducted using frequencies and proportion of young infants identified at different levels (CHW, nurse), proportion referred to hospital, proportion whose families did not accept the referral, proportion treated at the health centre, the proportion able to complete treatment, adherence to treatment rates, deaths and other adverse outcomes, the proportion that did not improve and needed higher-level care, the proportion that was cured and proportion that survived, along with information about barriers and facilitating factors.

**Implementation outcomes.** Feasibility of the intervention to increase coverage, quality of the implementation, and identification of challenges and solutions during implementation.

**Quantitative outcomes.** *Primary outcome.* The proportion of infants identified with signs of PSBI receiving appropriate treatment was designated as the primary outcome to evaluate the effectiveness of the implementation strategy.

*Secondary outcomes.* Death within two weeks after initiation of treatment, clinical deterioration defined as the emergence of any sign of critical illness, a new sign of CSI, or and any serious adverse effect of the antibiotics used for treatment were designated as secondary outcomes. Treatment failure was a composite outcome of death, clinical deterioration, or signs signifying the inadequacy of the original treatment and/or the need to change treatment course.

**Ethical considerations.** The study protocol and all associated data collection instruments and consent forms were approved by the Ethics Review Committee of WHO (MCA00415 dated 14 August 2015) and the Kinshasa School of Public Health Institutional Review Board (ESP/CE/083/2015 of 7 August 2015). Written informed consent was obtained for participation in the study, home visits for pregnancy and birth, enrolment and treatment, as well as for follow-up visits. This study was registered in the Australian New Zealand Clinical Trials Registry (ANZCTR), ACTRN12617001373369).

## Results

From April 2016 to March 2017, 3162 births were recorded at the participating health areas, of which we identified 3050 live births. CHWs visited 2909 (95%) young infants on day 1, 2680 (88%) on day 3, and 2656 (87%) on day 7 after birth.

We identified 285 young infants who resided in the study area and who had signs of PSBI, instead of 305 expected cases during the study period, based on an estimated 10% incidence of PSBI among 3050 live births [6,19]. Of these, 256 young infants received treatment for PSBI, thus the coverage of treatment was 83.9% (256/305). During the study period, based on the parents' addresses, we also determined that an additional 426 young infants came from outside the study area, thus a total of 711 young infants up to 2 months of age were identified with signs of PSBI. Over half (365/711, 51.3%) of these infants were brought by family members to the health centres. Of 426 (60%) who required referral to a hospital, two-thirds (282/426, 66.2%) refused the referral advice (Table 3).

Pneumonia was identified in 285 infants; 90.1% received outpatient treatment at a health centre (Table 3), and 99.2% took at least 13 doses of oral amoxicillin (Table 4). Treatment failure was identified in 3.1% with no deaths (Table 4).

Severe pneumonia was identified in 141 infants, who were referred to a hospital, but 90.8% of families refused referral advice and accepted treatment at the health centre level (Table 3). All of them received at least 13 doses of oral amoxicillin. Treatment failure was reported in 7.8% of these infants with no deaths (Table 4).

CSI was identified in 233 infants who were referred to a hospital; 63.1% of families refused referral advice and accepted treatment at the health centres (Table 3). All completed two injections of gentamicin, and 97.9% received at least 13 doses of oral amoxicillin (Table 4). Among those treated as outpatients, 17.0% failed treatment, and one died (Table 4).

Health centre nurses classified 52 infants as critical illness, referred them to a hospital, and 86.5% of families accepted referral advice (Table 3). Seven infants whose families refused referral completed seven days of treatment with injectable gentamicin and injectable ampicillin. Treatment failure occurred in 28.6% of infants, and all survived (Table 4).

Among the 539 infants with PSBI who received outpatient treatment, 99.2% completed seven days of treatment, 8.3% failed treatment, and one died.

**Table 3. Identification and confirmation of infants with signs of possible serious bacterial infection by health centre nurses (n = 711).**

| Parameters | Pneumonia (N = 285) n (%) | Severe pneumonia (N = 141) n (%) | Clinical severe infection‡ (N = 233) n (%) | Critical illness (N = 52) n (%) |
|---|---|---|---|---|
| **Care seeking** | | | | |
| Brought by families to a health centre | 210 (73.7) | 28 (19.9) | 108 (46.4) | 19 (36.5) |
| Identified by CHWs and referred or brought to a health centre | 75* (26.3) | 113* (80.1) | 125* (53.6) | 33 (63.5) |
| **Referral and treatment** | | | | |
| Treated at a health centre without referral to a hospital | 257† (90.1) | Not applicable | Not applicable | Not applicable |
| Those who needed referral and families accepted referral advice to a hospital | Not applicable | 13 (9.2) | 86 (36.9) | 45 (86.5) |
| Those who needed referral but families refused referral advice and accepted treatment at a health centre | Not applicable | 128 (90.8) | 147 (63.1) | 7 (13.5) |
| **Follow-up of those treated on an outpatient basis** | | | | |
| Day 4 follow-up completed | 256 (99.6) | 128 (100) | 147 (100) | 7 (100) |
| Day 8 follow-up completed‡ | 255 (99.2) | 128 (100) | 145 (98.6) | 7 (100) |

* CHWs identified one infant with pneumonia, one with severe pneumonia and two with signs of CSI; all diagnoses were confirmed by the health centre nurse during the home visit but refused to come to the health centre for treatment.

† 28 infants with pneumonia refused treatment at a health centre.

‡ denominator is all infants who received outpatient treatment in each category.

Among infants who were hospitalized, 7.7% with severe pneumonia died, 23.3% with signs of CSI died, and 40.0% with any sign of critical illness died within two weeks of the initial assessment. The overall case fatality ratio (CFR) among hospitalized infants with PSBI was 27%.

## Quality assurance

Before starting implementation, all study nurses, health centre nurses and CHWs were certified through standardization exercises, by passing with a minimum score of 85%. Subsequently, quality assurance exercises were conducted quarterly, and the accuracy of data completion was above 80%. During approximately 200 spot checks for gentamicin and 250 spot checks for amoxicillin, supervisors found no inappropriate dosing.

## Challenges and solutions during implementation

We encountered several challenges related to the weak health system and governance during implementation (Table 5). We implemented the PSBI intervention using the existing

**Table 4. Treatment outcomes among sick young infants who received outpatient treatment at health centres.**

| Parameters | Pneumonia (N = 257) n (%) | Severe pneumonia (N = 128) n (%) | Clinical severe infection (N = 147) n (%) | Critical illness (N = 7) n (%) |
|---|---|---|---|---|
| Completed treatment* | 255 (99.2) | 128 (100) | 144 (97.9) | 7 (100) |
| Treatment outcome | | | | |
| Treatment success | 247 (96.1) | 118 (92.2) | 119 (81) | 5 (71.4) |
| Treatment failure | 8 (3.1) | 10 (7.8) | 25 (17) | 2 (28.6) |
| Death | 0 (0) | 0 (0) | 1 (0.7) | 0 (0) |
| Lost to follow-up | 0 (0) | 0 (0) | 2 (1.3) | 0 (0) |

* Completed treatment for pneumonia or severe pneumonia was 13–14 doses of oral amoxicillin; for CSI two injections of gentamicin and 13–14 doses of oral amoxicillin; for critically ill a daily injection of gentamicin plus a twice-daily injection of ampicillin.

**Table 5. Challenges and solutions during implementation.**

| Challenges | Solutions |
|---|---|
| Facility staff and Community Health Workers (CHWs) lacked motivation as they were underpaid, and CHWs were expected to work for free | Small monetary incentives were given to both nurses and CHWs by the study team to maintain the fidelity of the intervention |
| Lack of essential medicines and supplies | Medicines were purchased and supplied to facilities using the study funds. Timers to count respiratory rate, thermometers and weighing scales were provided by the Management Sciences for Health funded by United States Agency for International Development |
| Although services were supposed to be free at the primary level, the families were frequently asked to pay the costs of care | As part of the study, all medicines were offered free of cost to all patients with PSBI |
| Lack of transportation that made the referral to the hospital difficult | CHWs were given bicycles and used their bicycles to transport families |

infrastructure and health care force to ensure the smooth implementation of the programme and to respond to the needs of the community. We used study funds to overcome the main challenges. The challenges were regular topics for discussion during the meetings between the TSU and local health authorities.

## Discussion

We demonstrated the feasibility of the successful implementation of the WHO PSBI management guideline [21] in a programme setting during a one-year study period. All five health centres in the study area were able to provide outpatient treatment of PSBI. The coverage of treatment was 84% at either the first level facility health centre or a hospital. Nearly, all of the sick young infants treated on an outpatient basis received adequate treatment according to the WHO guideline [8,21]. Those young infants whose families were unable to accept referral advice were treated with the WHO recommended simplified antibiotic regimens [21] at the health centres on an outpatient level with high treatment success and low mortality rates. Young infants 7–59 days of age with only fast breathing represented two-fifths of all sick young infants and were treated on an outpatient basis without referral to a hospital.

In general, the nurses at the health centres were able to correctly assess, classify and manage sick infants with PSBI. They were able to correctly follow the national guideline for management, including the provision of injectable antibiotics when the referral was not feasible. There was only one death amongst all the sick young infants with PSBI signs treated on an outpatient basis, suggesting that treatment was effective. Also, spot checks for antibiotic dosage did not find any problems in our study, whereas 23% errors in dose calculation and 15% classification errors on assessment were reported from the MaMoni project, Bangladesh [24]. The training that health centre staff received and the oversight and handholding by supervisors and the TSU staff contributed to reinforcing the skills of the health workers and building their confidence. The critical role of technical support, mentoring and supervision of health care providers, has also been reported previously from PSBI implementation research studies in the MaMoni project and Kushtia, Bangladesh, Ntcheu, Malawi and Zaria, Nigeria [10,11,25,26].

In our study, treatment adherence was very high in those treated on an outpatient basis. Nearly all young infants treated on an outpatient basis completed their treatment. Such high rates have also been reported from other PSBI implementation research sites [10,11,13,26]. This may have resulted from the high-quality training, supervision and motivation of health

workers as well as the acceptance of simplified treatment on an outpatient basis by the families [13]. Whereas, in India [12], the adherence rates were lower than those seen in our study.

During the study period, we identified over 3000 live births, and we expected that around 10% of young infants would develop signs of PSBI [6,19,20]. The number of young infants living in the study area who were identified with PSBI represented about 9% of live births, which was close to the published data and suggested that almost all potential cases of PSBI were identified and brought to the health centres for appropriate treatment. Our coverage of treatment was a bit less than that reported from Zaria, Nigeria (96%) [10]; but higher than those reported from Kushtia, Bangladesh (31%) [26]; Lucknow, India (53%) [12]; and Ntcheu, Malawi (64%) [11]. However, in our study, the overall number of PSBI patients were much higher than expected during the study period. Review of the addresses showed that only two-fifths of these sick infants were from the study area, whereas the majority were from the neighbouring villages. Families living near the participating health centres and even from further away found out about the free essential medicines and other supplies through word of mouth and came to seek care at these facilities.

Two-third of families whose sick young infants needed hospital care refused referral advice and accepted simplified antibiotic treatment on an outpatient basis. This is nearer to the median of 74% (range 34–97%) reported in the systematic review of community-to-facility neonatal referral completion rates in Africa and Asia [9]. Other PSBI implementation research sites have also reported high referral refusal rates i.e., 97% in Zaria, Nigeria [10]; 93% in Ntcheu, Malawi [11]; 81% in Lucknow, India [12]; and 83% in Sylhet and Lakshmipur, Bangladesh [13].

We observed 27% CFR among those who received treatment in a hospital compared to 0.2% CFR among those treated at the health centres. Higher CFR among infants receiving hospital care is likely to have resulted from greater severity of illness among this cohort. Alternate or additional explanations for higher CFR among infants receiving hospital care include delays in reaching the hospital due to the family decision-making process, distance to the hospital, lack of suitable transport, or poor quality of care at the hospital. The latter may be due to poor infrastructure, or lack of essential equipment and medicines at district hospitals in the study area. Both referral hospitals lacked oxygen, incubators and blood bank services, and sometimes essential medicines. Their laboratories are not equipped to perform cultures. In comparison, another PSBI implementation research site from Lucknow, India, reported 13% CFR among those treated at a hospital [12], compared to 2.6% CFR among those treated on an outpatient basis at the government health centres by trained health workers, whereas CFR was 25% among those sick young infants whose families refused treatment at any government facility and sought only private treatment.

In our study, we observed a higher treatment failure rate of 17% in young infants with signs of CSI compared to 5% reported in the AFRINEST severe infection study [19], which may be explained by the real-life setting of our study which was different from a controlled trial. Other PSBI implementation research studies have reported variable but generally, low treatment failure rates in CSI cases i.e., 3.5% in Malawi [11], 1.3% in Zaria, Nigeria [10]; and 5.8% in Lucknow, India [12]. However, our treatment failure rate of 3% for infants with pneumonia was lower than 19% reported in the AFRINEST pneumonia trial [20]. But, the AFRINEST study treatment failure definition also included persistence of fast breathing on day 4, which if excluded would result in a failure rate of 3.7% [20]. Other PSBI implementation research studies have reported variable but low treatment failure rates in fast breathing pneumonia cases i.e., none in Zaria, Nigeria [10]; 2.2% in Malawi [11], and 4.8% in Lucknow, India [12].

There are several programme implications of our results. First, although we have demonstrated that it is possible to implement the WHO guideline for PSBI treatment [21] using

existing infrastructure, additional investments were needed to ensure success. Health workers were provided with quality training, incentives and additional supervision, and facilities were supplied with essential medicines. Families were receptive to advice by CHWs on home care, including referral when needed, when they knew that quality services were available and affordable. We believe that the feasibility of implementation depends upon the provision of these resources. Second, we demonstrated to the national and local health authorities that it is possible to provide simplified antibiotic regimens to sick infants with PSBI at the primary level of care, thereby, contributing to a reduction in young infant deaths in the DRC. However, for this intervention to be part of the overall national strategy for the elimination of preventable deaths, investments must be made in strengthening the health system. Third, the success of using a model with oversight and consultative support provided by a consortium of government and non-governmental technical experts (i.e. the TSU) suggests the importance of the partnership between technical experts and programme implementers. The TSU played an important role in capacity-building and problem-solving, which may have provided programme staff with confidence for implementation. The TSU participated in planning, training/capacity strengthening, supervision and monitoring, quality control and problem-solving aspects of the study. To scale up this intervention, some technical support to the health system in the future would be needed. Technical members of the TSU may also have benefitted from their role by gaining an understanding of the potential contributions they can make to build a responsive and resilient health system.

The key lessons learned from this implementation research that may be of use to policymakers include the value of government commitment and substantial external support and partnership. Uninterrupted delivery of essential commodities is critical for maintaining the quality of services and confidence of communities. Technical support, mentoring and supervision are important to sustain the quality of care of services. The need for health system strengthening for achieving and sustaining high coverage of PSBI treatment is essential. Poor quality of care at hospitals may result in low acceptability of referral advice and preference for outpatient treatment.

Our study had some strengths. First, the MOH played a vital role in the implementation of the study. MOH staff from central, district and health zone levels were actively engaged in the study and were looking forward to the eventual integration of the findings into the routine health system. This strong support is crucial not only for generating community buy-in and ownership but also for promoting quality assurance of the intervention. Second, this study was implemented in a programme setting. People who work in the real world (health centre nurses) were used to assess infants with danger signs and provide appropriate treatment. Although support to acquire various commodities, including antibiotics, was provided through study funding, all medicines were purchased from the regional distribution centre, used by all government health facilities for the purchase of medicines and health supplies. The existing health system monitoring, evaluation and supervision system was used and strengthened, thus ensuring the sustainability of the process. Finally, community buy-in was promoted as part of the implementation. Village leaders sensitized the community for timely care-seeking and facilitated referral by CHWs.

Our study also had some limitations. Although we demonstrated that community-based identification and treatment of PSBI was feasible, we cannot necessarily infer feasibility and sustainability on a larger scale because of some aspects of our strategy. First, we had to fill gaps in areas of usual responsibility of the district and national health authorities. Staff salaries and provision of essential medicines and supplies are core functions of any government, and unless these are addressed sustainably, it will be difficult for the DRC to scale up this strategy and make progress towards the achievement of the Sustainable Development Goals [27]. The DRC

has committed to universal health coverage and needs to take lessons learned from this implementation research to identify sustainable solutions for scaling-up. Second, CHWs are village volunteers and, as such, are not entitled to payment. However, in this study, CHWs received a small monetary incentive and bicycles. We do not know what their performance might have been without this incentive. Also, health centre nurses received a small monetary incentive. This may have generated extra motivation that would not exist under scale-up conditions. Third, almost all critical supplies (thermometers, respiratory rate timers, weighing scales, medicines) were provided by the study and an international agency. The success of scale-up would not be possible without this level of assistance. Finally, due to limited resources, we could not increase the capacity of the district hospitals. We collected limited outcome data from hospitalized patients, but couldn't collect detailed clinical data.

In conclusion, our study provides policy-makers and programme implementers with strategies for addressing the management of sick young infants with PSBI when the referral is not feasible. Implementing the PSBI guideline in a programme setting is feasible but will require appropriate investments and external support. Treatment protocols can be standardized, and trained health care providers can manage sick young infants with simplified antibiotic regimens, which in turn will save lives. Health system challenges need to be addressed to successfully implement the strategy. Community sensitization and improvements in quality of care at the referral level are complementary strategies that the MOH needs to strengthen to prevent deaths and improve survival. This implementation research has identified critical areas for action and has shown the roadmap for innovative strategies for addressing the needs of sick young infants and for leaving no one behind in the context of universal health coverage.

## Supporting information

**S1 Table. Dose and frequency of oral amoxicillin.**
(PDF)

**S2 Table. Dose and frequency of injectable antibiotics.**
(PDF)

## Author Contributions

**Conceptualization:** Adrien Lokangaka, Samira Aboubaker, Shamim Ahmad Qazi, Rajiv Bahl.

**Formal analysis:** Adrien Lokangaka, Daniel Ishoso, Antoinette Tshefu, Michel Kalonji, Samira Aboubaker, Shamim Ahmad Qazi, Yasir Bin Nisar, Rajiv Bahl, Carl Bose, Yves Coppieters.

**Methodology:** Adrien Lokangaka, Samira Aboubaker, Shamim Ahmad Qazi, Rajiv Bahl, Carl Bose.

**Project administration:** Adrien Lokangaka, Daniel Ishoso, Antoinette Tshefu, Michel Kalonji, Paulin Takoy, John Otomba.

**Supervision:** Adrien Lokangaka, Daniel Ishoso, Michel Kalonji, Paulin Takoy, John Otomba, Samira Aboubaker, Shamim Ahmad Qazi, Rajiv Bahl.

**Writing – original draft:** Adrien Lokangaka, Daniel Ishoso, Antoinette Tshefu, Samira Aboubaker, Shamim Ahmad Qazi, Yasir Bin Nisar, Rajiv Bahl, Carl Bose, Yves Coppieters.

**Writing – review & editing:** Adrien Lokangaka, Daniel Ishoso, Antoinette Tshefu, Jack Kokolomami, Samira Aboubaker, Shamim Ahmad Qazi, Yasir Bin Nisar, Rajiv Bahl, Carl Bose, Yves Coppieters.

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
