## [Decision Letter · Decision Letter 0]

24 Sep 2021

PONE-D-20-38722

Simplified antibiotic regimens for young infants with possible serious bacterial infection when referral is not feasible in the Democratic Republic of the Congo

PLOS ONE

Dear Dr. Lokangaka,

Thank you for submitting your manuscript to PLOS ONE. After careful consideration, we feel that it has merit but does not fully meet PLOS ONE’s publication criteria as it currently stands. Therefore, we invite you to submit a revised version of the manuscript that addresses the points raised during the review process.

We look forward to receiving your revised manuscript.

Kind regards,

Mehreen Arshad, M.D.

Academic Editor

PLOS ONE

Additional Editor Comments (if provided):

Thank you for submitting this very interesting manuscript. The reviewers, while mostly appreciative of the data presented, had some minor comments noted below.

Journal Requirements:

The authors declare no conflict of interest. Rajiv Bahl and Yasir Bin Nisar are staff members of the World Health Organization. The expressed views and opinions do not necessarily express the policies of the World Health Organization

6. Please upload a copy of Supporting Information Supplementary Table 1 and 2 which you refer to in your text on page 28.

Reviewers' comments:

Reviewer's Responses to Questions

**Comments to the Author**

1. Is the manuscript technically sound, and do the data support the conclusions?

Reviewer #1: Yes

Reviewer #2: Yes

Reviewer #3: Yes

Reviewer #4: Yes

Reviewer #5: Yes

2. Has the statistical analysis been performed appropriately and rigorously? 

Reviewer #1: Yes

Reviewer #2: Yes

Reviewer #3: Yes

Reviewer #4: N/A

Reviewer #5: Yes

3. Have the authors made all data underlying the findings in their manuscript fully available?

Reviewer #1: Yes

Reviewer #2: Yes

Reviewer #3: Yes

Reviewer #4: Yes

Reviewer #5: Yes

4. Is the manuscript presented in an intelligible fashion and written in standard English?

Reviewer #1: Yes

Reviewer #2: Yes

Reviewer #3: Yes

Reviewer #4: Yes

Reviewer #5: Yes

5. Review Comments to the Author

Reviewer #1: The authors have highlighted an important topic considering the problem of limited health care access to underprivileged population in the region. The manuscript has written in detail. However, I am unable to grab information on the Qualitative component of the project.

Methods:

Comment 1: They have mentioned that it was the mixed method study but how the study was conducted whether qualitative component conducted first or the Quantitative component ? i.e. is it the sequential mixed method study design? please clarify this in the methodology

It seems that during implementation of intervention they have conducted FGDS and KII and table 4 were their results but how many FGDs and KII conducted? this information is missing

Results:

Comment 2: This component is too long, authors have described each and everything in the text. it can be shortened by mentioning only key information in the text and for other variables it is better to direct the reader to the table.

Comment 3: There is discrepancy in proportions in text and tables. Refer to table 2 on page 17 and text in lines 385 and 394 on page 18

Tables:

Comment 4: if operational definitions of pneumonia, severe pneumonia and CSI or critical illness are mentioned in the methods then it can be removed from table 2 and 3 footnotes.

Reviewer #2: Thank you for inviting me to review the manuscript.

The manuscript is well written and provides interesting findings on the treatment of PSBI in young infants in outpatient settings. However, some clarifications are needed.

1- It may be important to precise the proportion of neonatal infections due to bacterial infections worldwide and/or in LMIC/DRC to appreciate the importance of the implementation of outpatient treatment for possible serious bacterial infections in young infants.

2-The authors did not indicate the proportion of patients identified by CHW in the study area. This information could be useful to evaluate their importance as with the current data this might be underestimated.

3- If we assume that the CHW work only in their health areas, all patients from outside the study area should be brought by their family to the health centers. So how could we have 426 patients from outside the study area and only 385 walk-in patients?

A total of total of 285 young infants with PSBI were identified in the study area and 346 were identified by the CHW. Did the CHW and/or nurses from other health areas also referred patients to health centers in the study area?

4- The authors have identified some challenges to the implementation of their protocol and have adopted certain solutions at the beginning of the study. However, there is no mention of the appropriateness and/or evaluation of these solutions: any delay in the transportation of patients due to unavailability of HCW or insufficient number of bikes (which might have influenced the outcome of patients treated at the district hospital)? Stock-outs of medications despite the expected higher number of patients?

5- The mortality in the group of participants treated in the referred hospital was higher compared to those treated at home or at the health centers. Moreover, this mortality is higher compared to previous studies. The authors have provided some interesting explanations to these findings; however, could the difference between the two groups also explain these results? For example, the distribution of comorbidities in the two groups, families with sicker patients more likely to accept referral.

6-Line 108: please define the acronym “PSBI” on its first occurrence in the text.

Reviewer #3: The manuscript is easy to understand, very descriptive, informative and generally well-written. Minor changes mainly relating to grammatical errors are required. The introduction should also be expanded to provide comprehensive information on other factors that may induce serious bacterial infections in infants such as the mother`s immunity and neonatal immunisation.

Reviewer #4: In the Abstract introduction, the author mentioned that injectable antibiotics may not be feasible in many low resource settings. Is there any reason, e.g access? Please indicate.

The author should add the WHO recommendation in the introduction and not just reference it.

Reviewer #5: Thank you for the opportunity to review this manuscript. The implications of which are important to improving child mortality especially in Africa. A few minor changes are suggested:

- The author could include a subheading in the methods with study definitions to describe what is meant by pneumonia, severe pneumonia, critical illness, presumed serious bacterial infection, young infant age ranges used, adequate treatment, etc. This may eliminate having to provide the definitions under each table in the article

- The age range of a young infant is included at every mention throughout the document which need not be so if defined at the beginning of the article

Abstract:

- This a mixed methods longitudinal study but in the results, no mention is made of the salient findings from the qualitative side of the study

Introduction:

- Line 99: should be “per 1000 live births”

-

Data Analysis:

- lines 343-347: was treatment failure (with the same/similar criteria as the secondary outcomes) a separate secondary outcome?

Results:

- As this was a mixed method intervention study, what were the findings from the interviews/focus groups of the families?

- What were the reasons given by parents for refusal of transfer to a health facility?

- lines 361-364 reword the sentence to simplify it

- lines 381 -397: Table 2 and 3 summarise your results of the primary and secondary outcomes so there is no need to mention all of the findings in the text. The day 4 and day 8 follow up would imply that a certain number of antibiotic doses were taken. This could be cumulatively summarized for the clinical syndromes analysed.

- Provide the overall case fatality rate of those infants hospitalised

Discussion:

- lines 447-448: “whereas from 23% errors in dose calculation and 15% classification 448 errors on assessment were reported from MaMoni project, Bangladesh” remove the word “from”

- Line 438: remove the reference in the middle of the statement, it has been referenced from the previous statement

- Lines 456-457: the other research sites have been mentioned in the previous paragraph, there is no need to re-list them if referenced

6. PLOS authors have the option to publish the peer review history of their article (what does this mean?). If published, this will include your full peer review and any attached files.

Reviewer #1: **Yes: **Sonia Qureshi

Reviewer #2: No

Reviewer #3: No

Reviewer #4: **Yes: **Nigel Aminake Makoah

Reviewer #5: No

---

## [Author Response · Author response to Decision Letter 0]

5 Nov 2021

I have attached a rebuttal letter giving point by point responses to reviewers' and editor's comments.

---

## [Decision Letter · Decision Letter 1]

29 Mar 2022

PONE-D-20-38722R1

Simplified antibiotic regimens for young infants with possible serious bacterial infection when the referral is not feasible in the Democratic Republic of the Congo

PLOS ONE

Dear Adrien Lokangaka

Thank you for submitting your manuscript to PLOS ONE. After careful consideration, we are happy to accept your paper for publication conditional to your making  the minor revisions suggested by reviewers 1 and 6. Therefore, we invite you to submit a revised version of the manuscript that addresses the points raised during the review process.

We look forward to receiving your revised manuscript.

Kind regards,

Nusrat Homaira

Academic Editor

PLOS ONE

Journal Requirements:

Reviewers' comments:

Reviewer's Responses to Questions

**Comments to the Author**

1. If the authors have adequately addressed your comments raised in a previous round of review and you feel that this manuscript is now acceptable for publication, you may indicate that here to bypass the “Comments to the Author” section, enter your conflict of interest statement in the “Confidential to Editor” section, and submit your "Accept" recommendation.

Reviewer #1: All comments have been addressed

Reviewer #2: All comments have been addressed

Reviewer #3: All comments have been addressed

Reviewer #4: (No Response)

Reviewer #5: All comments have been addressed

Reviewer #6: All comments have been addressed

2. Is the manuscript technically sound, and do the data support the conclusions?

Reviewer #1: Yes

Reviewer #2: Yes

Reviewer #3: Yes

Reviewer #4: Yes

Reviewer #5: Yes

Reviewer #6: Yes

3. Has the statistical analysis been performed appropriately and rigorously? 

Reviewer #1: Yes

Reviewer #2: Yes

Reviewer #3: Yes

Reviewer #4: I Don't Know

Reviewer #5: Yes

Reviewer #6: Yes

4. Have the authors made all data underlying the findings in their manuscript fully available?

Reviewer #1: Yes

Reviewer #2: Yes

Reviewer #3: Yes

Reviewer #4: Yes

Reviewer #5: Yes

Reviewer #6: Yes

5. Is the manuscript presented in an intelligible fashion and written in standard English?

Reviewer #1: Yes

Reviewer #2: Yes

Reviewer #3: Yes

Reviewer #4: Yes

Reviewer #5: Yes

Reviewer #6: Yes

6. Review Comments to the Author

Reviewer #1: The authors have tried to address all the comments.

In methodology, it would be great if authors will describe and elaborate the qualitative component such as: how many FGDs and KII were conducted? What was the size of each FGD? Where FGDs were conducted etc.

Reviewer #2: Thank you for inviting me to review this paper. The authors have satisfactorily addressed my concerns.

Reviewer #3: The reviewed manuscript is entitled: “Simplified antibiotic regimens for young infants with possible serious bacterial infection when referral is not feasible in the Democratic Republic of the Congo”.

The manuscript is easy to understand, very descriptive, informative and generally well-written.

Reviewer #4: WHO recommendation on how much injectable antibiotics should be given was not added in the introduction, probably due to limitations on the number of words?

Reviewer #5: Thank you for the revised manuscript.

Some minor grammatical changes suggested:

Line 447: rather say " as an outpatient" or "on an outpatient basis"

Line 429: insert "and" between the two categories of infants in the sentence

Line 450: the reference number at the end of the sentence

Be consistent with the text format throughout the document especially Table 2

Reviewer #6: (No Response)

7. PLOS authors have the option to publish the peer review history of their article (what does this mean?). If published, this will include your full peer review and any attached files.

Reviewer #1: **Yes: **Sonia Qureshi

Reviewer #2: No

Reviewer #3: No

Reviewer #4: No

Reviewer #5: **Yes: **Harsha Lochan

Reviewer #6: No

---

## [Editor Report · Decision Letter 2]

27 Apr 2022

Simplified antibiotic regimens for young infants with possible serious bacterial infection when the referral is not feasible in the Democratic Republic of the Congo

PONE-D-20-38722R2

Dear Adrien Lokangaka

We’re pleased to inform you that your manuscript has been judged scientifically suitable for publication and will be formally accepted for publication once it meets all outstanding technical requirements.

Kind regards,

Nusrat Homaira

Guest Editor

PLOS ONE
---

## [Editor Report · Acceptance letter]

21 Jun 2022

PONE-D-20-38722R2 

Simplified antibiotic regimens for young infants with possible serious bacterial infection when the referral is not feasible in the Democratic Republic of the Congo 

Dear Dr. Lokangaka:

I'm pleased to inform you that your manuscript has been deemed suitable for publication in PLOS ONE. Congratulations! Your manuscript is now with our production department. 

Kind regards, 

on behalf of

Dr. Nusrat Homaira 

Guest Editor

PLOS ONE